# Analysis of the Influence of Microbial Community Structure on Flavor Composition of Jiang-Flavor Liquor in Different Batches of Pre-Pit Fermented Grains

**Shiming Shen** [1,2,3], **Jinlong Liu** [2], **Ruiqi Luo** [3], **Jiaojiao Zhang** [3], **Dong Zhao** [4], **Xinxin Xue** [3], **Jia Zheng** [4], **Zongwei Qiao** [4], **Qiang Zhang** [4], **Zheng Feng** [4] **and Xinglin Han** [1,3,*]

[1] Key Laboratory of Brewing Molecular Engineering of China Light Industry, Beijing Technology and Business University, Beijing 100048, China
[2] College of Food and Biology, Hebei University of Science and Technology, Shijiazhuang 050018, China
[3] China National Research Institute of Food and Fermentation Industries, International Joint Research Center of Quality and Safety of Alcoholic Beverages, Beijing 100015, China
[4] Yibin Wuliangye Group Limited, Yibin 644000, China
[*] Correspondence: hanian2009@163.com

**Abstract:** To explore the effects of microbial community changes on the key flavor substances in base Baijiu, this study analyzed the microbiome of the pre-pit fermentation grains using high-throughput sequencing technology and determined the flavor substances of the base Baijiu by GC-FID. The results showed the microbial community changed dynamically between the different rounds, as well as bacteria and fungi displayed different succession patterns. Next, the variations of skeletal flavor substances in the base Baijiu were analyzed by multiomics, and it was found that alcohols, acids, and esters were the most abundant, accounting for 88.16–98.87% of the total flavor substances, and decreased with the increase of the rounds. By calculating the Spearman coefficient, it was found that microorganisms such as Acinetobacter, Oceanobacillus, Saccharomyces, and Byssochlamys were significantly correlated with the n-Propano and 2,3-Butanediol and other components in the base Baijiu. Finally, 15 flavor substances such as Acetaldehyde, Propionaldehyde, and Isobutyraldehyde were identified as key substances by OAV analysis. This study is the first to reveal the potential association between the microbial community of pre-pit fermentation grains and flavor of base Baijiu and has the benefit of improving the quality of base Baijiu.

**Keywords:** Jiang-flavor Baijiu; pit fermentation; high-throughput sequencing

## 1. Introduction

Jiang-flavor Baijiu (JFB) is one of the typical flavors of traditional Chinese Baijiu, becoming popular for its elegant and delicate flavor, with the characteristics of obvious sauce flavor and lingering fragrance in empty cups [1]. JFB is also widely welcomed by Chinese people for its excellent quality and unique flavor and has gradually become popular all over the world. The flavor substances in JFB are complex, accounting for only 2% of the total content, mainly consisting of alcohols, acids, esters, pyrazines, aldehydes, and ketones, aromatics, and other compounds [2]. The flavor substances of JFB are closely related to its complex process and microbial community [3]. In brief, JFB production is a long time and complicated process, which requires two times of grains feeding, nine times of steaming, eight times of fermentation, and seven times of extractions, and finally the finished baijiu can be obtained after storage and blending. (Figure 1)

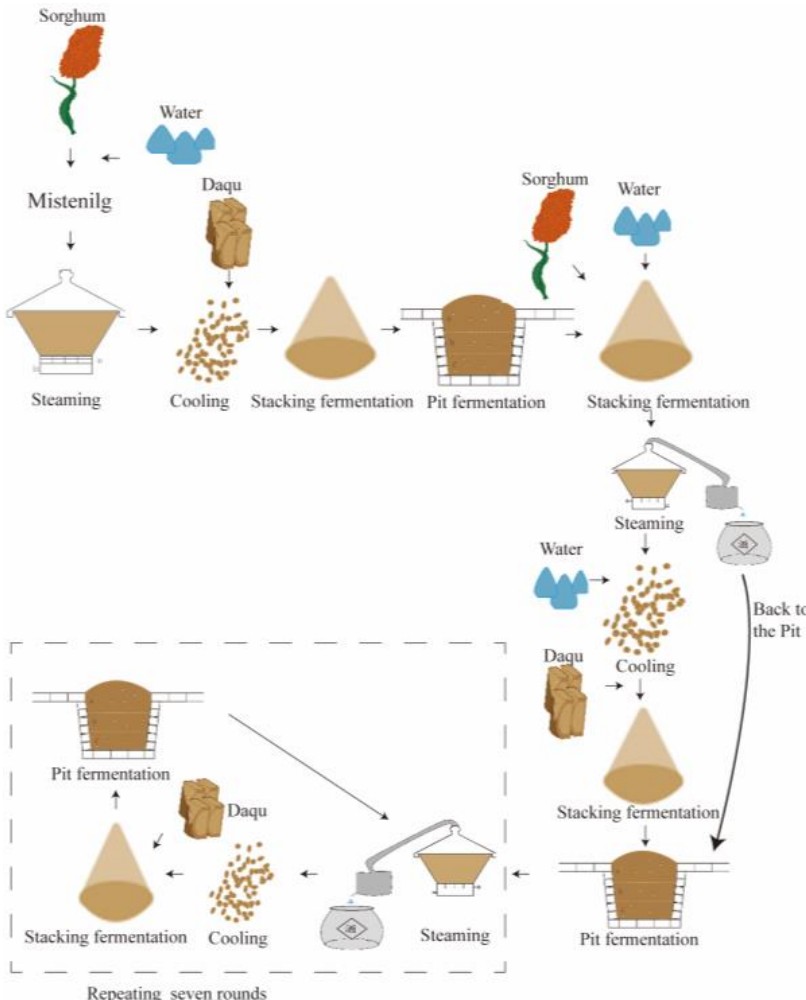

**Figure 1.** The process of Jiang-flavor Baijiu. The production period of JFB is one year. Firstly, sorghum is used as raw material to go through moistening and steaming, next high temperature Daqu (made of wheat, fermented, and stored) is used as starter, stirred and mixed well, then it undergoes 4–5 days of stacking fermentation and 30 days of pit fermentation, and finally the first round of baijiu is obtained through high temperature distillation. After steaming, distilled grains are mixed with Daqu, and the process of stacking fermentation and pit fermentation are repeated, resulting in a total of seven rounds of base Baijiu.

Microorganisms play an important role in producing alcohol, metabolizing flavor substances, saccharifying raw materials, and promoting fermentation, which have a significant impact on the flavor formation and quality of baijiu [4]. Influenced by temperature, oxygen, and other factors, the microbial community changes dynamically and forms a unique microbial community composition during the processes of distiller's yeast (Daqu) making, stacking, and pit fermentation [5]. Stacking is a generally used method of microbial enrichment in the field of fermentation and is traditionally used to promote flavor in JFB. However, its precise mechanism is unknown. High-throughput sequencing technology is a common method for analyzing the structure of microbial communities. Shi et al. used high-throughput sequencing technology to track the changes of microbial community during the production of Daqu, and the results showed that different spatial environments caused Daqu to exhibit variability in microbial composition and metabolism, and formed Daqu with different sensory characteristics (white, yellow, and black) [6]. Li et al. analyzed the microbial diversity and metabolites dynamic of light-flavor Baijiu with stacking process and found that dominant genera such as *Lactobacillus*, *Bacillus*, and *Pichia* were positively correlated with flavor compounds such as esters and acids [7]. The bacterial microbial

community structure of fermented grains in different rounds of Lutaichun JFB produced in north China was studied by high-throughput sequencing technology, and the potential probiotics strains with good characteristics (alcohol tolerance, etc.) were screened. Xiao et al. also analyzed the changes of microbial communities during the fermentation process of baijiu in the field and laboratory with the help of high-throughput sequencing technology, and the results demonstrated that the microbial community of Daqu exhibited species specificity and periodic succession with some underlying patterns [8]. Similarly, Wang et al. used high-throughput sequencing to analyze the changing patterns of microbial communities in different rounds of the stacking process and found that the diversity of microbial communities showed differences depending on fermentation parameters such as moisture and acidity, and there were both characteristics of microbes and shared core microbes in different rounds of the stacking process [9]. During the fermentation phase in the pit, the composition of the microbial community evolves further under the influence of the anaerobic environment, producing flavor substances that become part of liquor [10]. After stacking, the grains enter the pit to ferment and become the foundation for microbes, and then the microbes produce flavor substances in an anaerobic environment, becoming the flavor substances in the base liquor.

At present, researchers focused more on the relationship between microbial communities and flavor substances at the same stage [11], but there is a lack of research on the influence of the current microbial community on the flavor composition in the next stage. Stacking is one of the most important processes that affect the quality of JFB and is also the last process that can be easily regulated manually. During the stacking fermentation, grains were mixed with Daqu, and were saccharified and enriched with microorganisms in the environment, setting the foundation for pit fermentation [12]. After stacking, accumulated microorganisms enter the pit and produce flavor substances in an anaerobic environment, which become the flavor substances in the base Baijiu [13].

In this study, we examined the composition of the microbial community of the pre-pit fermentation grains by using high-throughput sequencing technology and characterized the core microorganisms, dominant microorganisms, and characteristic microorganisms in different rounds of pre-pit fermentation grains. The GC-MS technique was used to identify the flavor substances of the base Baijiu and to analyze the differences in flavor substances of the base Baijiu over different rounds. Finally, the correlation between microbial community of the pre-pit fermentation grains and the flavoring substances of the base Baijiu was investigated according to the partial least squares method, which deepened the understanding of the microbial community in pre-pit fermentation grains and provided the foundation for liquor production regulation.

## 2. Materials and Methods

### 2.1. Sample Collection

Samples were collected in Renhuai city, Guizhou province, China. JFB production is a long process, comprising of seven rounds, in which the flavor of raw Baijiu is different. All samples were collected from the same workshop of JFB fermentation and were divided into two types: pre-pit fermentation grains (PPF) and base Baijiu. Pre-pit fermentation grains samples were collected from the last day of stack fermentation in the nominated locations as shown in Figure 2. One pre-pit fermentation grain sample was formed by sampling five times (A, B, C, D, and E) at fixed sites and then mixing well. The post-pit fermentation grains were distilled out in layers, and alcohol was taken from fixed parts of each layer of grains three times (a, b, c), and then all the alcohol samples from the three layers were evenly mixed to form a sample of the base Baijiu to be tested. For convenience, all samples are renamed as shown in Table 1.

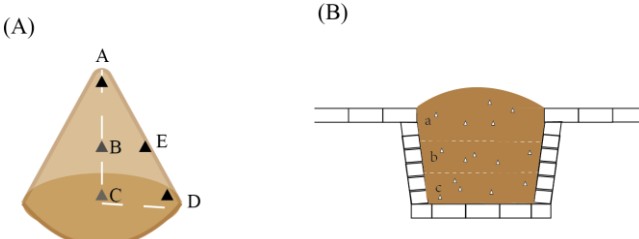

**Figure 2.** Schematic diagram of the sample collection. The sample sources (pre-pit fermentation grains and post-pit fermentation grains) are divided into different zones. (**A**) Sampling site for pre-pit fermentation grains A: top of the pile, B: center of the pile, C: bottom of the pile, D: bottom of the pile exterior, E: middle of the pile exterior. (**B**) Sampling site for post-pit fermentation grains. a: upper fermented grains, b: middle fermented grains, c: low fermented grains.

**Table 1.** Sample information and abbreviations.

| Samples | Information | Samples | Information |
|---------|-------------|---------|-------------|
| R1-PPF | pre-pit fermentation grains in round 1 | R1 | Base Baijiu in round 1 |
| R2-PPF | pre-pit fermentation grains in round 2 | R2 | Base Baijiu in round 2 |
| R3-PPF | pre-pit fermentation grains in round 3 | R3 | Base Baijiu in round 3 |
| R4-PPF | pre-pit fermentation grains in round 4 | R4 | Base Baijiu in round 4 |
| R5-PPF | pre-pit fermentation grains in round 5 | R5 | Base Baijiu in round 5 |
| R6-PPF | pre-pit fermentation grains in round 6 | R6 | Base Baijiu in round 6 |
| R7-PPF | pre-pit fermentation grains in round 7 | R7 | Base Baijiu in round 7 |

Note: Samples is sample abbreviation, Information is sample information.

### 2.2. Determination of Organic Acids in Base Baijiu

For the determination of organic acids in base Baijiu, ion chromatography was used; in order to reduce the effect of ethanol in the Baijiu on the column, the base Baijiu sample needs to be diluted: 200 μL of base Baijiu is first aspirated and then diluted 60 times using ultrapure water, followed by an external standard method for quantitative determination.

ICS-3000 ion chromatography analyzer (with EG40 automatic generator, conductivity detector and Chromeleon 6.80 chromatograph workstation): DIONEX, USA; Clarn 600 meteorological chromatograph (with flame ion detector): Perkin Elmer, USA; ion chromatography conditions: conductivity detector detection: drench rate: 1.0 mL/min; injection volume: 25 μL. KOH drenching program: 0–16 min, KOH concentration 1.1 mmol; 16–29 min, KOH concentration linearly varies from 1.1 to 16.5 mmol; 29–35 min, KOH concentration linearly varies from 16.5 to 20 mmol; 35–39 min, KOH concentration linearly varies from 20 to 35 mmol; 39–41 min, KOH concentration linearly varies from 20 to 35 mmol. ~35 mmol; 39–41 min, KOH concentration remained 35 mmol; 41–47 min, KOH concentration varies linearly from 35 to 50 mmol.

Organic acid standards (in peak order): lactic acid, acetic acid, propionic acid, formic acid, and n-butyric acid.

### 2.3. Determination of the Flavor Components of the Base Baijiu

The skeletal flavor of the base baijiu was determined by GC-FID, characterized by the external standard method and quantified by the internal standard method. A 10 mL sample of base Baijiu was taken, then 100 μL of 1% of the three internal standard solutions were added, mixed, and then injected into the sample for analysis. The internal standard solutions were prepared by pipetting 1 mL each of tert-amyl alcohol, 2-ethylbutyric acid and n-butyl acetate, respectively, and fixing them with 100% ethanol to 100 mL; the standard solutions were prepared by pipetting the corresponding volumes (mL) of the 46 target compounds into 100 mL volumetric flasks, weighing them on a 10,000 ppm balance, and fixing them with 100% ethanol to 100 mL. The working solutions were diluted step-by-step to form different concentrations (with the addition of the internal standard solution) and analyzed to establish the standard working curve.

GC-FID, autosampler, hydrogen flame ionisation detector (PerkinElmer, Waltham, MA, USA); capillary column CP-Wax 57CB, type: 50 m × 0.25 mm × 0.2 μm.

GC conditions: carrier gas (N2) flow rate 1 mL/min, hydrogen flow rate 45 mL/min; air flow rate ratio 450 mL/min; shunt ratio 10:1; injector temperature 240 °C; tester temperature 270 °C. Column temperature: 35 °C, 6 min, ramp to 60 °C at 4 °C/min, 110 °C at 6 °C/min, 3 min, 205 °C at 6 °C/min, 13 min. Injection volume: 1 μL.

Standards (in peak order): acetaldehyde, propionaldehyde, isobutyraldehyde, ethyl formate, ethyl acetate, acetal, 2-pentanone, sec-butanol, n-propanol, isobutanol, ethyl valerate, isoamyl acetate, n-butanol, activated pentanol, 2-pentyl alcohol, isoamyl alcohol, ethyl heptanoate, n-pentyl alcohol, acetic buzz, ethyl caproate, ethyl lactate, n-hexanol, isoamyl caproate, ethyl caprylate, furfural, 2,3-butanediol, ethyl nonanoate, ethyl decanoate, 1,2-propanediol, ethyl tetradecanoate, beta-phenylethanol, heptanoic acid, octanoic acid, ethyl palmitate, ethyl oleate, ethyl stearate, and ethyl linoleate.

The standards required for quantification were all imported reagents, mainly from Sigma-Aldrich, ACROSORGANICS and Fluka, with purity $\geqq$ 97%.

### 2.4. DNA Extraction and High-Throughput Sequencing of Microbial Community

Total DNA extraction: A total of 5 g of pre-pit fermentation grains sample was weighed and quickly ground in liquid nitrogen. The extraction method follows the operating instructions of the E.Z.N.A. Soid DNA Kit. PCR amplification: PCR amplification of the V3-V4 high variant region of the bacterial 16S r RNA gene was performed with primers 338F/806R (5′-ACTCCTACGGG-AGGCAGCA-3′/55′-GGACTACHVGGGTWTCTAAT-3′, and amplification of the internal transcribed spacer region (ITS1) of the fungus was performed with primers ITS5F/ITS1R (5′-GGAAGTAAAAGTCGTAACAAGG-3′/5′-GCTGCGTTCTTCAT CGATGC-3′), amplified and detected by 1% agarose gel electrophoresis. PCR amplification conditions: pre-denaturation at 94 °C for 3 min; denaturation at 94 °C for 30 s; annealing at 55 °C for 45 s, extension at 72 °C for 45 s, 30 cycles; and extension at 72 °C for another 10 min. Illumina Mi Seq sequencing: Libraries were constructed using Thermofisher's Ion Plus Fragment Library Kit 48 rxns library building kit. After the constructed libraries were qualified by Qubit quantification and library testing, they were sequenced using Thermofisher's Ion S5TMXL. Sequencing data processing: Using Cutadapt (V1.9.1, http://cutadapt.read-thedocs.io/en/stable/, accessed on 6 February 2021), the reads are first cut in the low-quality section, then the sample data are split from the obtained reads according to barcode, and the raw data are obtained by truncating the barcode and primer sequences for preliminary quality control. (Raw reads) The reads obtained after the above processing need to be processed to remove the chimeric sequences (http://www.drive5.com/usearch/manual/chimera_formation.html, accessed on 6 February 2021), and the reads sequences are processed by (UCHIME Algorithm, http://www.drive5.com/usearch/manual/chimera_formation.htmluchime_algorithm, accessed on 6 February 2021). (http://www.drive5.com/usearch/manual/uchime_algo.html, accessed on 6 February 2021) with the species annotation database to detect chimeric sequences, and finally remove the chimeric sequences to obtain the final valid data (clean reads).

### 2.5. Analysis of Data

All clean reads from all samples were clustered using Uparse software (Uparse v7.0.1001, http://www.drive5.com/uparse/, accessed on 6 February 2021), which by default clusters sequences into OTUs (operational taxonomic units) with 97% identity. The highest frequency of OTUs is selected as the representative sequence of OTUs according to the principles of the algorithm. Chao1, Shannon, Goods coverage indices were calculated using Qiime software (Version 1.9.1). The composition of the microbial community was calculated from the relative abundance of OTUs at the phylum and genus levels. Correlation analysis was studied using the pheatmap software package software and visualised using correlation heat maps, and changes in flavor substances in different rounds of base

Baijiu were analyzed using Venn diagrams and heat maps. SPSS 25.0 was used to calculate Spearman's correlation coefficients between microorganisms and base baijiu flavor substances, and Gephi (0.9.6) was used to visualize the analysis. Partial least squares regression analysis of microorganisms and key flavor substances in base baijiu was conducted using SIMCA 14.1.

## 3. Results

### 3.1. Microbial Community Diversity Analysis

The OTUs' (operational taxonomic units, OTU) coverage estimates of all bacteria and fungi were more than 99.5% in all samples, indicating that the sequencing results can truly reflect the abundance and diversity of microbes in all samples (Table 2). The variation of Shannon and chao1 indices suggested that there were differences in the microbial community of different rounds of the pre-pit fermentation grains (PPF), as well as bacteria and fungi which displayed different succession patterns.

**Table 2.** Analysis of microbial diversity in different rounds.

| Samples | Coverage | | Shannon | | Chao1 | |
|---|---|---|---|---|---|---|
| | **Bacteria** | **Fungi** | **Bacteria** | **Fungi** | **Bacteria** | **Fungi** |
| R1-PPF | 0.996 | 1.000 | 3.059 | 0.665 | 1026.252 | 62.000 |
| R2-PPF | 0.999 | 1.000 | 1.371 | 1.256 | 205.279 | 36.750 |
| R3-PPF | 1.000 | 1.000 | 4.942 | 3.391 | 993.754 | 192.937 |
| R4-PPF | 1.000 | 1.000 | 4.288 | 2.985 | 340.286 | 156.000 |
| R5-PPF | 0.999 | 1.000 | 2.836 | 0.463 | 704.439 | 15.000 |
| R6-PPF | 1.000 | 1.000 | 4.834 | 1.987 | 246.240 | 118.588 |
| R7-PPF | 0.999 | 1.000 | 4.787 | 0.854 | 341.652 | 46.333 |

Note: Coverage index: indicates the sequencing depth, the higher the value, the higher the coverage; Shannon index: indicates the colony diversity, the higher the value, the higher the colony diversity; Chao1 index: indicates the colony abundance, the higher the value, the more colony species.

### 3.2. Microbial Community Dynamic of Pre-Pit Fermentation Grains in Different Rounds

Analysis of the microbial community composition of the pre-pit fermentation grains in different rounds by high-throughput sequencing technology showed that *Firmicutes*, *Proteobacteria*, and *Actinobacteriota* were the dominant phyla in all seven rounds at the phylum level of bacteria, with comparative abundance all greater than 1% (Figure 3A). The comparative abundance of Firmicutes declined with increasing rounds (70.31–86.67% in round 1–4 and 7.41–18.08% in round 5–7). The comparative abundance of *Proteobacteria* and *Actinobacteriota* showed an increasing trend with increasing rounds. At the phylum level of fungi, *Ascomycota* was the dominant fungal phyla in all seven rounds, with comparative abundance percentages of 95.72–99.99% (Figure 3B).

Due to the complex and variable composition of the microbial community, microorganisms with comparative abundance greater than 0.1% were chosen for analysis at the genus level, including 17 bacterial genera and 16 fungal genera (Figure 3C,D). At the bacterial genus level, *Lactobacillus*, *unidentified_Mitochondria*, *Bacillus*, *Kroppenstedtia*, and *Oceanobacillus* were the common microorganisms in the seven rounds, among which *Bacillus*, *Kroppenstedtia*, and *Oceanobacillus* had higher comparative abundance in round 3–5, accounting for 2.42–8.39%, 12.45–21.25%, and 2.77–9.31%, respectively (Table S1). At the level of fungal genera, *Issatchenkia*, *Pichia*, and *Zygosaccharomyces* were the dominant fungal genera in the seven rounds, accounting for 68.34–99.99%, with *Zygosaccharomyces* having a higher comparative abundance of 2.66–55.11% in round 3–5. Classification of the microbiota into core genera (relative abundance > 20%) and dominant genera (comparative abundance 1–20%) was conducted according to comparative abundance [14,15]. The results showed that the core and dominant microorganisms of different rounds of the pre-pit fermentation grains are unique (Table 3). According to the records, the genus with greater than 1% abundance in a single round and significantly higher than that of other rounds was consid-

ered to be the characteristic genus of that round [9]. Thereby, in the bacterial community, *Pediococcus* and *Weissella* were the characteristic genus of the 1st round, *Thermoactinomyces* and *Virgibacillus* were the characteristic genus of the 3rd round, *Oceanobacillus* was the characteristic genus of the 4th round, and *Ralstonia* was the characteristic genus of the 6th round (Tables 3 and S1). In the fungal community, the core microorganisms were mainly *Issatchenkia* and *Pichia*. *Issatchenkia* was the common core microorganisms for round 1, 2, 5, 6, and 7, and *Pichia* was the common core microorganisms for round 2, 3, and 7. *Byssochlamys* and *Saccharomyces* were characteristic microorganisms of the 1st round of stacking, *Candida* of the 3rd round, *Kazachstania, Monographella, Thermoascus,* and *Trichocladium* of the 4th round and *Aspergillus, Wickerhamomyces,* and *Mucor* of the 6th round (Table S2). The results showed that composition of bacterial and fungal community was distinctive among the rounds of stacking and displayed variable dynamics during the fermentation of Baijiu.

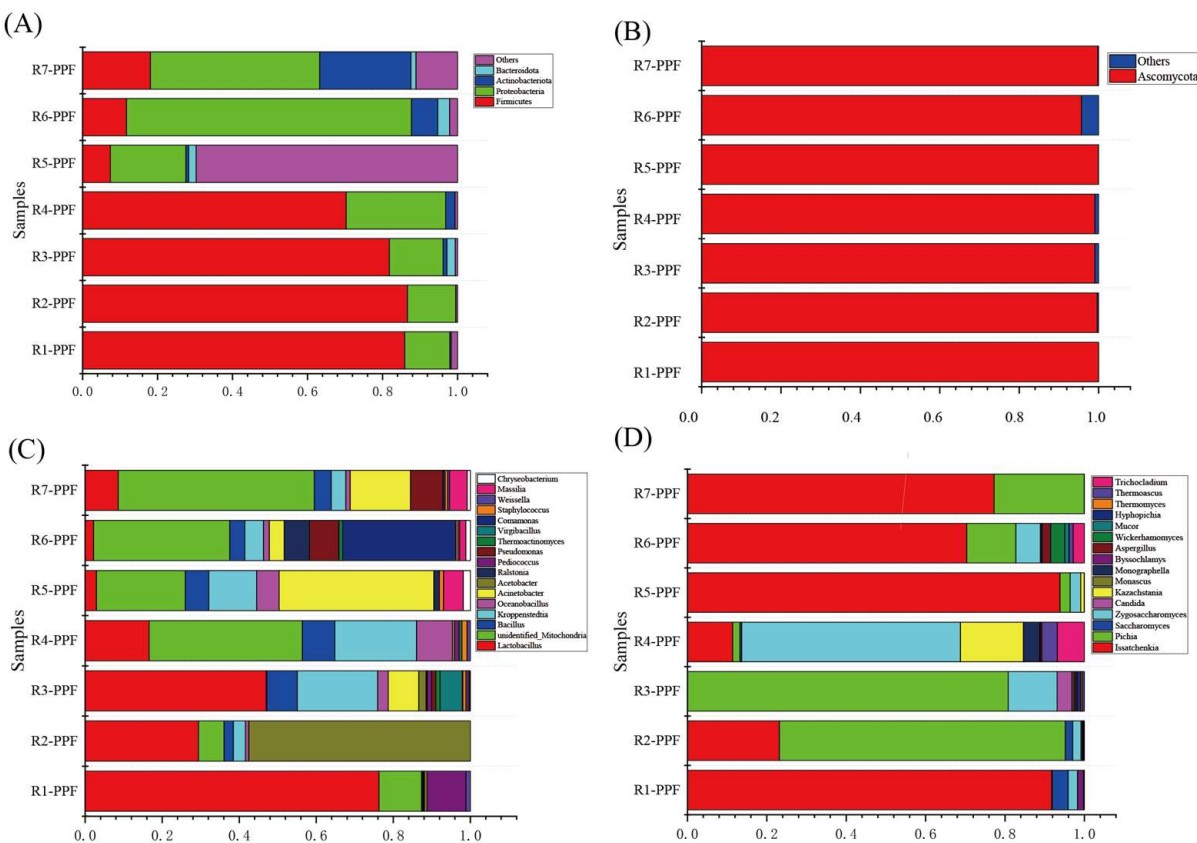

**Figure 3.** The composition of the microbial community over different rounds of the pre-pit fermentation grains. (**A**) Percent of bacterial community abundance on phylum level. (**B**) Percent of fungi community abundance on genus level. (**C**) Select bacterial genera with comparative abundance greater than 0.1% for the map, containing a total of 17 bacterial genera. (**D**) Select fungal genera with comparative abundance greater than 0.1% for the map, containing a total of 16 fungal genera.

**Table 3.** Core microbe, dominant microbe, and characteristic microbe during different rounds of pre-pit fermentation grains.

| Samples | Core Microbe | | Dominant Microbe | | Characteristic Microbe | |
|---|---|---|---|---|---|---|
| | Bacteria | Fungi | Bacteria | Fungi | Bacteria | Fungi |
| R1-PPF | *Lactobacillus* | *Issatchenkia* | *unidentified_Mitochondria,* | *Zygosaccharomyce* | *Pediococcus, Weissella* | *Byssochlamys, Saccharomyces* |
| R2-PPF | *Lactobacillus, Acetobacter* | *Issatchenkia, Pichia* | *unidentified_Mitochondria, Bacillus, Kroppenstedtia* | *Saccharomyces, Zygosaccharomyce* | | |
| R3-PPF | *Lactobacillus, Kroppenstedtia* | *Pichia* | *Acinetobacter, Acetobacter, Pediococcus, Pseudomonas* | *Zygosaccharomyce, Candida* | *Thermoactinomyces, Virgibacillus* | *Candida* |
| R4-PPF | *unidentified_Mitochondria, Kroppenstedtia* | *Zygosaccharomyces* | *Lactobacillus, Bacillus, Pediococcus, Staphylococcus* | *Issatchenkia, Pichia* | *Oceanobacillus* | *Kazachstania, Monographella, Thermoascus, Trichocladium* |
| R5-PPF | *unidentified_Mitochondria, Acinetobacter* | *Issatchenkia* | *Ralstonia, Staphylococcus, Massilia, Chryseobacterium* | *Pichia, Zygosaccharomyce* | | |
| R6-PPF | *unidentified_Mitochondria, Comamonas* | *Issatchenkia* | *Acinetobacter, Pseudomonas, Massilia, Chryseobacterium* | *Pichia, Zygosaccharomyce, Thermoascus, Trichocladium* | *Ralstonia, Comamonas* | *Aspergillus, Wickerhamomyces, Mucor* |
| R7-PPF | *unidentified_Mitochondria* | *Issatchenkia, Pichia* | *Acinetobacter, Massilia* | | | |

*3.3. Analysis of Flavor Substances in Different Rounds of Base Baijiu*

The composition of Jiang-flavor Baijiu (JFB) is complex and contains thousands of flavor substances, which are mainly divided into skeletal components and trace components [16]. Among them, the skeleton components determine the aroma of the JFB, while the trace components determine the flavor and typicality of the JFB [17]. According to some research studies, this study determined 51 important flavor substances in JFB using ion chromatography analyzer and GC-FID. The results showed that these 51 flavor substances were higher in the base Baijiu, among which 40 flavor substances were common to the seven rounds of base Baijiu (Figure 4A), which were the basic flavor substances constituting different rounds of base Baijiu and had an important influence on the quality of the base Baijiu. Moreover, with the progress of fermentation, the total content of flavor substances in the base Baijiu gradually decreased from 36.992 g/L (R1) to 9.605 g/L (R7) from the 1st round to the 7th round, in which the content of alcohols was most obviously decreased from 27.112 g/L (1st round) to 2.136 g/L (7th round) (Figure 4B, Table S3).

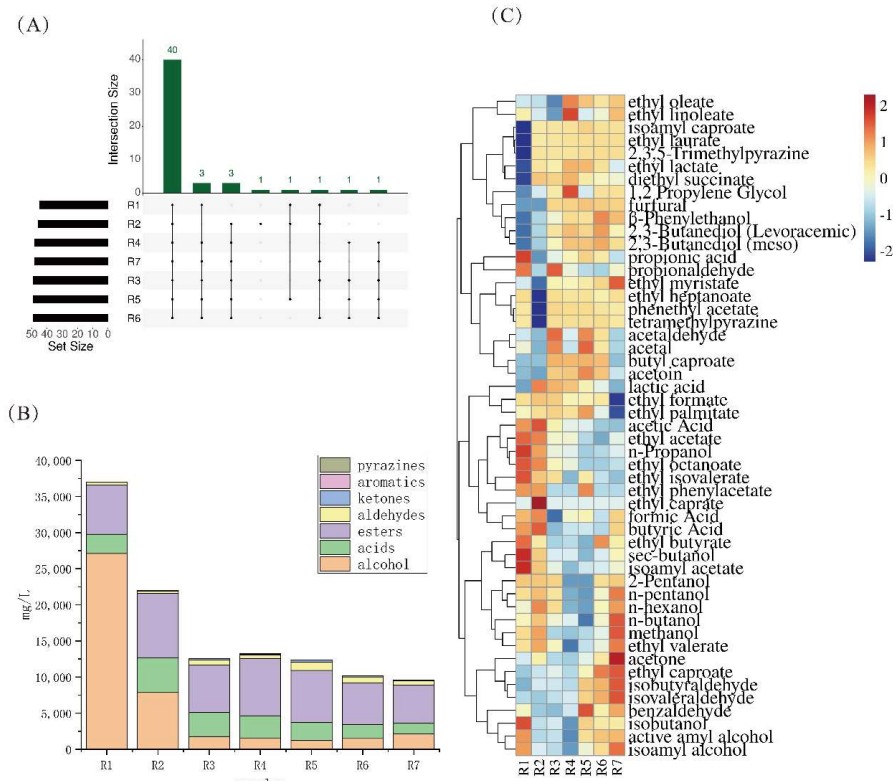

**Figure 4.** Variation in the flavor content in different rounds of the base Baijiu. (**A**) Advanced Venn diagram analysis of flavor substances in seven rounds. (**B**) Analysis of the composition of flavor substances in base Baijiu. (**C**) Clustering heat map analysis of the change pattern of flavor substances.

Of the 51 flavor substances, there were 13 alcohols, 5 acids, 21 esters, 2 pyrazines, 6 aldehydes, 2 ketones, and 2 aromatics. Among them, alcohols, acids, and esters were the three most abundant flavor substances in the seven rounds of base Baijiu, accounting for 88.16–98.87% of the total flavor substance content (Table S3). Overall, it seems that alcohols had the highest relative content in the first round of base Baijiu, including methanol, n-propanol (fruity and grassy aroma), sec-butanol (fruity aroma), and isobutanol (wheat aroma), which was beneficial to the fruity and grains aroma of the base Baijiu. The second round of base Baijiu had relatively the highest content of acids, containing lactic acid, acetic acid (vinegar flavor), and butyric acid (sour flavor); therefore, the second round of base Baijiu had a strong sour flavor. The higher proportion of esters was in round 2–7, which contain ethyl oleate, ethyl linoleate, ethyl lactate (green, floral and fruity), and butyl caproate; so, round 2–7 base Baijiu had lower alcohol content and stronger

aroma [18]. Aldehydes and ketones, on the other hand, accounted for a higher percentage in round 3–5, giving the base Baijiu a rich aroma and prominent sauce. The aromatic and pyrazine substances were less in the base Baijiu, and their content increased slightly with the increasing rounds, which promoted the aroma harmony of the base Baijiu. The flavor substance composition of the base Baijiu changed from round to round, resulting in changes in the aroma and taste of the base Baijiu (Figure 4C). Further analysis of the variation of the content of each flavor substance between different rounds showed that the content of n-propanol and ethyl acetate decreased with the increase of rounds, ranging from 25.000 to 0.642 g/L and 5.338 to 2.506 g/L, respectively; the contents of 2,3-butanediol (levulinic), 2,3-butanediol (endocyclic), ethyl caproate, ethyl laurate, and isovaleraldehyde increased with increasing rounds, with isovaleraldehyde content ranging from 0.023 to 0.092 g/L, while the relative contents of methanol, sec-butanol, isoamyl alcohol, formic acid, butyric acid, and ethyl butyrate were lower in round 3–5 than in other rounds, and the relative contents of 1,2-propanediol, ethyl lactate, ethyl palmitate, ethyl oleate, acetaldehyde, acetal, and furfural in rounds 3–5 were higher than those in other rounds, among which ethyl lactate had the highest relative percentage, ranging from 3.532 to 4.987 g/L.

### 3.4. Correlation between Microbial Communities of Pre-Pit Fermentation Grains and Flavor Substances of Base Baijiu in Different Rounds

To explore the influence of changes of the microbial communities of pre-pit fermentation grains on the flavor of the base Baijiu, this study constructed a network correlation diagram between microbial communities and base Baijiu flavor substances based on the Spearman correlation coefficient. In the network diagram for bacteria covering 15 species of bacteria and 38 kinds of flavor substances, there were 99 pairs of significant relationships, including 43 pairs of positive and 56 pairs of negative correlations. In the network diagram for fungi, which included 13 fungi and 37 flavor substances, a total of 53 pairs of significant relationships were identified, including 17 pairs of positive and 36 pairs of negative correlations. In the two network diagrams, the bacterial network diagram had more nodes and edges than the fungal network, and the average degrees of the two were 1.904 and 1.082, respectively, which indicated that the composition of both bacterial and fungal communities of pre-pit fermentation grains affected the production of flavor substances in the base Baijiu, but the bacterial community had a greater impact. The modularity indices for both the bacterial and fungal network diagrams were higher than 0.44, at 0.529 and 0.731, respectively, indicating that the fungal network of pre-pit fermentation grains had a higher degree of modularity and a stronger structure.

Analysis of the effect of bacteria on the flavor substances of the base Baijiu showed that among the eight alcohols, n-Propano was positively correlated with *Acinetobacter* and *Acetobacter*, while 2,3-Butanediol (Levoracemic) was positively correlated with *Oceanobacillus*, and the rest of the alcohols were negatively correlated with bacteria. Of all the three acids, acetic acid was positively correlated with *Acetobacter* and the remaining acids were negatively correlated with bacteria. Of all the 16 esters, 10 esters such as ethyl formate, ethyl acetate, and ethyl isovalerate were significantly positively correlated with specific microorganisms, with acetic acid and *Acetobacter* showing a highly significant positive correlation, ethyl acetate and *Lactobacillus* showing a highly significant positive correlation, isoamyl caproate and *Chryseobacterium* showing a highly significant positive correlation, and the remaining esters showing a negative correlation with bacteria. Of the other 10 flavoring substances, 2,3,5-Trimethylpyrazine was positively correlated with *Chryseobacterium* and *Tetramethylpyrazine*, acetal and acetoin were positively correlated with *Ralstonia*, ethyl caproate and furfural were positively correlated with *Chryseobacterium*, and acetoin was positively correlated with *Oceanobacillus* (Figure 5A).

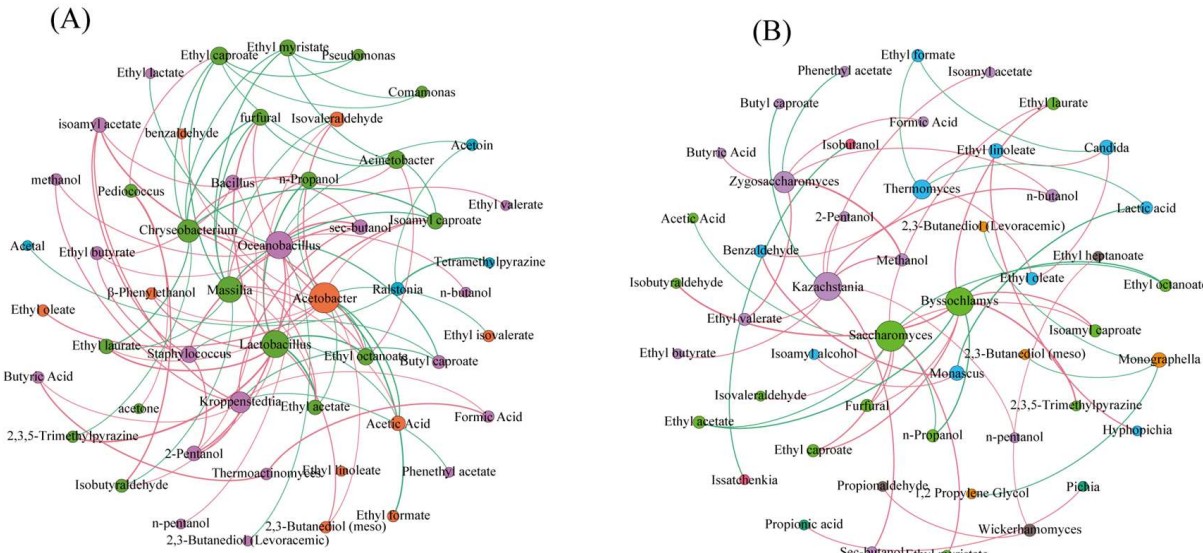

**Figure 5.** Correlation analysis of stacked microbial and base Baijiu flavor substances based on Spearman's correlation coefficient. (**A**) Correlation between bacteria and flavor substances. (**B**) Correlation between fungi and flavor substances. The size of the circles is linked to the number of the edge of compounds. The thickness of lines is proportional to the value of Spearman's correlation ($|\rho| > 0.6$, $p < 0.05$). Red and green lines indicate negative and positive correlations between microbial and flavor substances.

Analysis of the effect of fungi on the flavor substances of the base Baijiu revealed that of the 10 alcohols, n-Propanol was positively correlated with *Saccharomyces* and *Byssochlamys*, 1,2 Propylene Glycol, 2,3-Butanediol (Levoracemic), and 2,3-Butanediol (meso) were positively correlated with *Monographella*, and the remaining 6 alcohols were negatively correlated with fungi. Of the five acids, lactic acid was positively correlated with *Thermomyces* and *Monascus*, acetic acid was positively correlated with *Saccharomyces*, and the remaining three acids were negatively correlated with fungi. Of the 15 esters, ethyl formate was positively correlated with *Candida* and *Thermomyces*, ethyl acetate was positively correlated with *Saccharomyces* and *Byssochlamys*, butyl caproate was positively correlated with *Kazachstania*, ethyl octanoate was positively correlated with *Saccharomyces*, phenethyl acetate was positively correlated with *Zygosaccharomyces*, and the rest of the esters were negatively correlated with fungi; the rest of the pyrazines, aldehydes, and aromatics were negatively correlated with fungi (Figure 5B). Overall, the bacteria that had the greatest impact on the flavor profile of the base Baijiu were *Lactobacillus*, *Massilia*, *Oceanobacillus*, and *Acetobacter*; the fungi that had the greatest impact on the flavor profile of the base Baijiu were *Kazachstania*, *Saccharomyces*, *Byssochlamys*, and *Thermomyces*.

### 3.5. Regression Analysis of Microbial of Pre-Pit Fermentation Grains and Key Flavor Substances in Different Rounds

Further analysis of the relationship between the microbial community of the pre-pit fermentation grains and the key flavor substances in the base Baijiu was performed to provide a reference for regulating the key flavor substances of the base Baijiu. According to the perceived thresholds of 51 flavor substances based on some research studies [19], their OAV values were calculated [20], of which 40 flavor substances had OAV values greater than 1, and 15 flavor substances had OAV values greater than 100, which had obvious contributions to the flavor of JFB (Table 4).

**Table 4.** The 15 flavor compounds and aroma in different rounds of base Baijiu.

| Number | Compound | Characteristics | OAV |
|---|---|---|---|
| 1 | Acetaldehyde | Spicy, pungent odour | 339.40–696.28 |
| 2 | Propionaldehyde | Green Aroma | 168.33–658.97 |
| 3 | Isobutyraldehyde | Floral, fruity aroma | 1005.28–2926.81 |
| 4 | Isovaleraldehyde | Floral, fruity aroma | 1367.65–5416.53 |
| 5 | Acetal | fruity aroma | 6807.28–14,685.58 |
| 6 | Furfural | Almond aroma, sweet aroma, burnt, bitter | 0–4543.78 |
| 7 | isoamyl acetate | Banana, sweet, apple and fruit sugar aroma | 19.95–177.93 |
| 8 | Ethyl butyrate | Fruity, floral | 319.27–715.52 |
| 9 | Ethyl isovalerate | Apple, pineapple, banana and fruit aroma | 1058.13–7547.29 |
| 10 | Ethyl caproate | Sweet, fruity, pit and cucumber aroma | 1057.82–1897.14 |
| 11 | Ethyl octanoate | Pear, lychee, fruit, sweet, lily of the valley | 133.31–6904.38 |
| 12 | 1-Propanol | Fruity, floral and grassy aroma | 5.20–462.96 |
| 13 | 2,3-Butanediol (Levoracemic) | Sweetness | 129.17–381.94 |
| 14 | isoamyl alcohol | Fruity, floral aroma | 166.39–412.74 |
| 15 | Butyric Acid | Stench of sweat, sourness and pit mud | 77.33–691.23 |

The core microorganisms of the seven rounds were distributed in four quadrants, and the characteristic microorganisms of the same round were distributed in the same quadrant (Figure 6). In round 1 and 2, the core microorganisms (*Lactobacillus*, *Acetobacter*, and *Pichia*) and characteristic microorganisms (*Byssochiamys*, *Saccharomyces*, *Pediococcus*, and *Weissella*) were located in the 2nd quadrant. *Lactobacillus*, *Weissella*, *Acetobacter*, and *Pichia* of the pre-pit fermentation grains were potentially related to the production of Propionaldchyde, making the base Baijiu of round 1 and 2 have a greenish aromatic character. The dominant and characteristic microorganisms of round 3 and 4 were distributed in the 1st quadrant. *Bacillus* and *Staphylococcus* regulated the production of 2,3-Butanediol, which gave the base Baijiu its mellow sweetness. *Trichocladium*, *Thermoascus*, *Kazachstania*, *Monographella*, *Thermoactinomyces*, and *Zygosaccharomyces* regulated the production of acetaldehyde and acetal, resulting in a spicy and fruity base Baijiu. The core microorganism *Issatchenkia* was found in the 3rd quadrant and regulated the production of ethyl butyrate and isoamyl alcohol, which gave the base Baijiu its fruity and floral aroma. Round 5–7 had dominant and characteristic microorganism distributions (*Ralstonia*, *Comamonas*, *Aspergillus*, *Wickerhamomyces*, and *Mucor*) in the 4th quadrant. Characteristic microorganisms of round 6 regulated the production of glycolaldehyde, giving the base Baijiu of round 6 an almondy, burnt, sweet, and bitter aroma. The pre-pit fermentation grains core microorganisms (unidentified_Mitochondria) regulated the production of isobutyraldehyde, isovaleraldehyde, and ethyl caproate, giving the base Baijiu its fruity, floral, sweet, pit, and cucumber aroma.

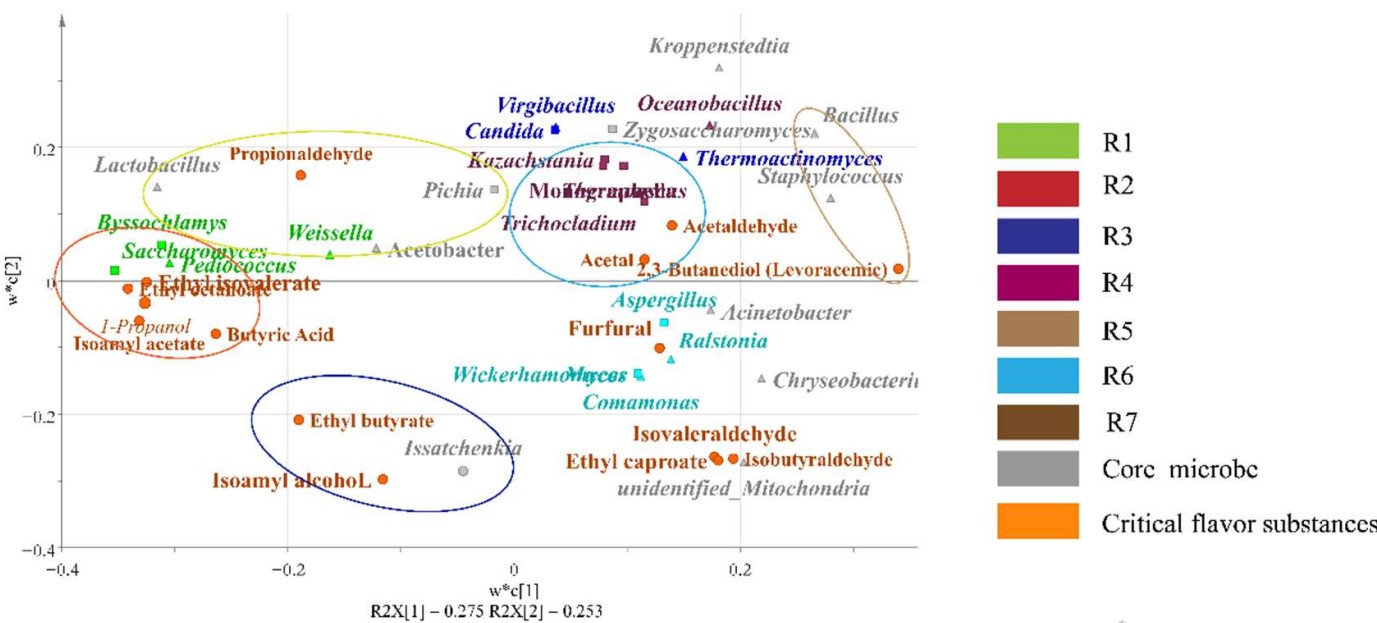

**Figure 6.** Analysis of the microorganisms of the pre-pit fermentation grains in different rounds and key flavor substances using partial least squares regression methods. (Triangles represent bacteria, squares represent fungi, and orange circles represent key flavor substances; seven colors represent seven rounds: R1–R7).

## 4. Discussion

The long-term experience of production showed that the fermentation of traditional Chinese Baijiu was semi-controllable and empirical [1].The key to achieve higher production requirements was to gain a deeper understanding of the growth and metabolic patterns of microorganisms, the mechanisms of flavor formation, and the factors that play an important role in the process [21]. In order to achieve these goals, each process in the production needs to be studied, which includes the use for raw materials, production and use of Daqu, stacking process, pit fermentation process, steaming process, storage process, and blending process. Much research has been published in relation to these processes; however, they were not well understanding of the potential association relationship between the changes in microbial communities of pre-pit fermentation grains and the flavor substances of the base Baijiu in different rounds. Therefore, this study investigated the diversity of microorganisms of pre-pit fermentation grains in different rounds by high-throughput sequencing, combined with GC-FID and ion chromatography analyzer to determine flavor substances and organic acids in base Baijiu, and finally analyzed the potential association between changes in microbial communities and flavor substances in base Baijiu.

The results showed that the composition of the microbial community of pre-pit fermentation grains in different rounds were specific, which made differences in the flavor of base Baijiu in different rounds and affected the quality of Baijiu. The composition of the microbial community of pre-pit fermentation grains had a close relationship with the Daqu. Related studies had also shown the specificity of the microbial communities of Daqu under different spaces and furthermore the formation of different types of Daqu (white, yellow, and black Daqu) [5,6]. Our study showed that *Firmicutes*, *Proteobacteria*, and *Actinobacteriota* were the dominant bacterial phylum and *Ascomycota* was the dominant fungal phylum before fermenting in the pit, which is consistent with previously reported findings. Among them, *Firmicutes* and *Ascomycetes* dominated not only the microbiota of post-stacking, but also the microbial communities of the Daqu, Jiupei, yellow water, and production environment, making an important contribution to the fermentation of Jiang-flavored Baijiu [22–24].

For the convenience of analysis, at the genus level, the microorganisms were classified into three categories according to their relative abundance: core microorganisms, dominant microorganisms, and characteristic microorganisms. Among them, *Lactobacillus*, *Acetobacter*, *Kroppenstedtia*, and *Comamonas* were the main core genera of bacteria before fermenting in the pit. Of these, *Kroppenstedtia* was also a core genus of black Daqu, with a much higher relative abundance than that of white and yellow Daqu [6].The relative abundance of *Bacillus* and *Oceanobacillus* (4th round characteristic genus) was higher in the white Daqu [9]. *Virgibacillus* had the highest relative abundance after the 3rd round in the pit and was more abundant in yellow than in white and black Daqu.

*Issatchenkia, Pichia,* and *Zygosaccharomyces* were the main core genus of fungi in the pit before fermenting. Of these, the characteristic genera *Byssochlamys* (round 1), *Thermoascus* (round 4), and *Aspergillus* (round 6) were in higher relative abundance in black Daqu and *Thermomyces* in white Daqu [25]. These results suggested that the composition of the microbial community in the pit may be related to the different types of Daqu added, and that the amount of black Daqu added may be higher. In related studies, *Pichia* (2nd round dominant genus) and *Thermoascus* (4th round characteristic genus) were the core microbes in the production environment, while *Wickerhamomyces* (6th round characteristic genus) was different significantly between production environments [25]. These studies suggested that the composition of the microbial community in the pit may be potentially linked to microbes in the production environment and possibly influenced by temperature and air pressure. By exploring the relationship between the composition of the microbiota in the pit and the microorganisms in Daqu and production environment, it helps to deepen the understanding of microbial community succession patterns in liquor production, and thus contribute to the regulation of liquor production.

A total of 51 flavor substances were examined from the base Baijiu in the seven rounds and 15 key flavor substances (OAV > 100) were identified based on activity intensity. Ethyl caproate, ethyl isovalerate, ethyl butyrate, butyric acid, and glyoxal also had OAV values above 100 in the study by Du et al. [26]. The concentration and ratio of ethyl caproate and ethyl butyrate have an important influence on the flavor of Baijiu, and the right amount of ethyl butyrate gives the Baijiu a clean and fresh taste [26]. In some views, ethyl acetate, ethyl lactate, ethyl butyrate, and ethyl caproate are the key esters in Baijiu and can reduce side effects after drinking [27]. Ethyl caprylate is also an important aroma compound and makes an important contribution to some of the aroma attributes of Baijiu [28].The content of n-propanol, 2,3-Butanediol (levulinic), and isoamyl alcohol in Baijiu also has an important influence on the aroma and is related to the activity of the amylase enzyme in the maceration [29]. In this study, n-propanol was potentially associated with *Saccharomyces*, which is a core microbe in the environment, so the level of n-propanol may be related to the microbial composition of the environment. The results of this study suggest a potential link between 2,3-Butanediol and *Staphylococcus*. In related studies, 2,3-Butanediol was found to be more abundant in black Daqu and the relative abundance of *Kroppenstedtia, Byssochlamys, Thermoascus*, and *Aspergillus* in black Daqu was higher [30]. Therefore, we hypothesize that these microbes in black Daqu may have interactions with *Staphylococcus* that ultimately regulate the production of 2,3-Butanediol. The key flavor substance butyric acid is one of the most frequently detected acids in Baijiu and also has high aroma vitality values in the literature [31]. The results of this study suggested that butyric acid was potentially associated with the characteristic genus of first round (*Byssochlamys, Saccharomyces, Pediococcus*, and *Weissella*). In the report, butyric acid was mainly produced by the genus *Clostridium*. Therefore, we speculated that *Clostridium* may interact with the characteristic genera of first round to ultimately regulate butyric acid content and contribute to the earthy odor of Jiang-flavor Baijiu [31].

## 5. Conclusions

The composition of flavor substances in base Baijiu is potentially linked to the composition of microbial communities at different stages, and the pre-fermented grains in

the pit are the last process in which microorganisms can be detected and regulated [32]. Therefore, we first analyzed the changes of microbial communities in seven rounds of PPF by high-throughput sequencing and determined 51 flavor substances in the base Baijiu. The results showed that the diversity of the microbial community changed in different rounds during the fermentation of liquor, and each had its own specificity. Further analysis revealed that the composition of the microbial community of pre-pit fermented grains may be related to the type and amount of the Daqu used and influenced by microorganisms in the environment. In addition, we found that changes in the pre-pit microbial community were potentially associated with key flavor substances in the base Baijiu, and these results could provide new insights into the regulation of flavor substance composition in the base Baijiu and provide a reference for further analysis of the intrinsic link between microorganisms and flavor substances.

**Supplementary Materials:** The following supporting information can be downloaded at: https://www.mdpi.com/article/10.3390/fermentation8120671/s1, Table S1: Relative abundance of bacteria in different rounds of grains. Table S2: Relative abundance of fungi in different rounds of grains. Table S3: The relative content of flavor substances in the base wine of the seven rounds.

**Author Contributions:** S.S. performed the main experiments and prepared figures and tables; J.L., R.L., D.Z., J.Z. (Jia Zheng), Z.Q., Q.Z. and Z.F. contributed the experiment materials and performed some experiments; S.S., R.L., J.Z. (Jiaojiao Zhang) and X.X. wrote the manuscript text; X.H. designed the experiments. All authors reviewed the manuscript. All authors have read and agreed to the published version of the manuscript.

**Funding:** This work was funded by the National Natural Science Foundation of China (Grant No. 32060532), the National Key Research and Development Program (Grant No. 2021YFE0192000), and the China National Research Institute of Food and Fermentation Industries.

**Institutional Review Board Statement:** Not applicable.

**Informed Consent Statement:** Not applicable.

**Data Availability Statement:** The raw reads of 16S rRNAs were deposited into the NCBI Sequence Read Archive (SRA) database (Accession Number: R1-PPF (SRR22138008), R2-PPF (SRR22138210), R4-PPF (SRR22188206), R5-PPF (SRR22188314), R6-PPF (SRR22188314), R7-PPF (SRR22188591)). The BioProject accession number: PRJNA896768 (R1-PPF), PRJNA896858 (R2-PPF), PRJNA898131 (R4-PPF), PRJNA898139 (R5-PPF), PRJNA898143 (R6-PPF), PRJNA898143 (R7-PPF). The BioSample accession number: SAMN31566272 (R1-PPF), SAMN31566701 (R2-PPF), SAMN31601327 (R4-PPF), SAMN31601597 (R5-PPF), SAMN31601654 (R6-PPF), SAMN31601679 (R7-PPF). The raw reads of ITS were deposited into the NCBI Sequence Read Archive (SRA) database (Accession Number: R1-PPF (SRR22188710), R2-PPF (SRR22188709), R4-PPF (SRR22188708), R5-PPF (SRR22188707), R6-PPF (SRR22188706), R7-PPF (SRR22188705)). The BioProject accession number was PRJNA898250, and BioSample accession number: R1-PPF (SAMN31601833), R2-PPF (SAMN31601834), R4-PPF (SAMN31601835), R5-PPF (SAMN31601836), R6-PPF (SAMN31601837), R7-PPF (SAMN31601838).

**Acknowledgments:** The anthors gratefully acknowledge Zhiwei Sun, Jinhui Hu, Mengchao He, Yuqi Chen and Wenmei Zhao from China National Research Institute of Food and Fermentation Industries (Beijing, China) for providing suggestions for this experimental study, and gratefully acknowledge Jinlong Liu and Ling Yang from Hebei University of Science and Technology (Shijiazhang, China) for providing suggestions for this experimental study.

**Conflicts of Interest:** The authors declare no conflict of interest. The authors declare that this study received funding from China National Research Institute of Food and Fermentation Industries. The funder was not involved in the study design, collection, analysis, interpretation of data, the writing of this article, or the decision to submit it for publication.

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
