# Peer review of "Analysis of the Influence of Microbial Community Structure on Flavor Composition of Jiang-Flavor Liquor in Different Batches of Pre-Pit Fermented Grains"

_fermentation, doi:10.3390/fermentation8120671_

Round 1

Reviewer 1 Report

In this study, the microbial communities of pre-pit fermentation grains were analyzed by high-throughput sequencing technology and the flavor substances of the base Baijiu were studied by GC-FID, which is of certain significance for the practical production of Jiang-flavor Baijiu. There are some minor issues in the article and the specific recommendations are as follows.

1. In the abstract section, there are some linguistic errors, such as "base wine", please check if there are similar errors in the whole article. In addition, there should be research implications or conclusion at the end of the abstract, please supplement it.

2. The current state of research in the introduction section appears to be more about daqu, which is not very relevant to this study, please revise it.

3. For lack of basis for the content of lines 227-228, please add references.

4. It is recommended that the results of the analysis of differential microorganisms (lines 231-236) be provided in the supplementary material.

5. In section 3.4, the relationship between microorganisms and volatile flavor substances is assessed by the topological properties of the co-occurrence network, so does the relationship between microorganisms in the network and the relationship between flavor substances affect the topological properties of the co-occurrence network?

6. streamline the illustrations in Figure 4 and Figure 5.

7."unidentified_Mitochondria" appears in line 402 and Figure 5, while "unidentified_Mitochondria" is not a microorganism and should be excluded from the analysis.

Author Response

Dear Reviewer,

Thank you for reviewing the paper, we have revised the manuscript based on your suggestions, as detailed in the response.  In case of any questions or any information you need, please feel free to contact me. I look forward to hearing from you.

Mr. Shiming Shen, First Author

Reviewer 2 Report

Revealing the dynamic changes of microbial communities and flavour compounds during pre-pit fermentation rounds are important, however, the manuscript should be improved.

Major concerns:

1. For the amplicon sequencing, no biological replicates were performed for each pre-pit fermentation round.

2. It seems that the HTS sequencing data was not deposited in public database such as NCBI or DDBJ or EBI.

3. The sequence of mitochondria should be removed before data processinig since the sequences of mitochondria were derived from grains rather than bacteria.

3. The accuracy of the absolute quantification data should be reconsidered. For example, the concentration of n-propanol in R1 and R2 was 25000 mg/L and 6747.13 mg/L (supplementary data), respectively. The concentration of this alcohol was too high to be believed. Please provide reference to support that the concent of propanol is credible.

4. The OAV and the correlation analysis can not provide usefull information because the OAV is related with the sensory of human.  Instead, the correlation analysis between concentation of flavour compopund and microbes would be better.

Minor concerns:

1. The resolution of figures were too low.

2. Numerous spelling and fonts mistakes.

3. Line numbers were missing.

Author Response

(The authors gave the same response as above.)

Round 2

Reviewer 2 Report

The authors have revised my concerns.